# KEQING: KNOWLEDGE-BASED QUESTION ANSWERING IS A NATURE CHAIN-OF-THOUGHT MENTOR OF LLMS

## ABSTRACT

Large language models (LLMs) have exhibited remarkable performance on various natural language processing (NLP) tasks, especially for question answering. However, in the face of problems beyond the scope of knowledge, these LLMs tend to talk nonsense with a straight face, where the potential solution could be incorporating an Information Retrieval (IR) module and generating response based on these retrieved knowledge. In this paper, we present a novel framework to assist LLMs, such as ChatGPT, to retrieve question-related structured information on the knowledge graph, and demonstrate that *K*nowledg*e*-based *q*uestion answer*ing* (***Keqing***) could be a nature Chain-of-Thought (CoT) mentor to guide the LLM to sequentially find the answer entities of a complex question through interpretable logical chains. Specifically, the workflow of ***Keqing*** will execute decomposing a complex question according to predefined templates, retrieving candidate entities on knowledge graph, reasoning answers of sub-questions, and finally generating response with reasoning paths, which greatly improves the reliability of LLM's response. The experimental results on KBQA datasets show that ***Keqing*** can achieve competitive performance and illustrate the logic of answering each question.

## 1 INTRODUCTION

Large language models (LLMs) (Brown et al., 2020; Chen et al., 2021a; Scao et al., 2022; Chowdhery et al., 2022; Zhao et al., 2023) have recently become the new darling of academia and industry due to their remarkable performance in a variety of natural language processing (NLP) tasks. With the blessing of techniques such as large-scale pre-training (Abnar et al., 2021), instruction tuning (Wang et al., 2022), and reinforcement learning from human feedback (RLHF) (Ziegler et al., 2019; Ouyang et al., 2022), existing pretrained LLMs have demonstrated unique capabilities in language understanding, generation, interaction, and reasoning. These powerful capabilities of LLMs also drive many emergent research topics (e.g., instruction learning (Wei et al., 2021), in-context learning (Brown et al., 2020), chain-of-thought prompting (Wei et al., 2022), etc.) to further investigate their huge potentials, and bring unlimited possibilities for humans to build advanced artificial intelligence systems. However, alongside these advancements, a pressing issue that plagues LLMs has been widely criticized as "*hallucination*", referred to as a phenomenon where LLMs tend to generate text that is incorrect, nonsensical, or not real (McKenna et al., 2023).

To alleviate the phenomenon of "*hallucination*" during the generation of LLMs, a promising direction is to retrieve the factual knowledge that are highly relevant to the user query, and then guide LLMs to generate response according to the retrieved context, resulting in retrieval-augmented LMs (Mialon et al., 2023; Oguz et al., 2020) that have recently demonstrated strong performance in knowledge intensive tasks, especially for knowledge-based question answering (KBQA). The workflow of existing retrieval-augmented LMs (Li et al., 2023a; Ram et al., 2023) mainly relies on embedding-based retrieval methods, which will first encode various forms of knowledge base and also the user query into the same latent space, then use a semantic similarity metric to retrieve the top-K most relevant documents as prompt, and finally instruct LLMs to only use the provided context to answer the user query. Due to the fact that embedding-based corpus retrieval often brings redundant context input, where repeated or irrelevant content will occupy a large number of tokens in the prompt, influencing the quality of response generated by LLMs (Li et al., 2023a). To alleviate this issue, we propose to construct a retrieval module operating on the knowledge graph to collecct relevant triplets, which can precisely provide high-quality context to assistant LLMs to complete the task of KBQA.

Distinct from previous KBQA methods (Cheng et al., 2022; Li et al., 2023a; Iyer et al., 2022) that utilize the reasoning capability of LLMs to directly generate a symbolic logical chain in SQL form to solve the user question, which is usually unexecutable in practice, in this work, we propose to use a *Question Decomposition* module to first decompose the complex user question into several sub-questions according to predefined question templates, where each question template can be solved with the pre-collected logical chains on knowledge graph, leading to several reasoning paths to reach the answer candidates to solve the user question. The thought behind such a design is that the logic of decomposing questions in text form could be easier to be captured by LLMs than that in SQL form, and for each real-world question, there could be multiple solutions (reasoning paths) to achieve the potential answer candidates, while sufficient answer candidates can provide tolerance for the following procedure of answer reasoning. After question decomposition and knowledge retrieval, with the retrieved answer candidates in hand, we utilize a *Candidate Reasoning* module to select the correct entities to answer the current question and iteratively result into the final answers according to the dependency of decomposed sub-questions.

Under the context of KBQA, the logical chains on the knowledge graph can naturally form chain-of-thoughts (CoT) (Wei et al., 2022) to guide existing LLMs to decompose complex questions into several sub-questions, which is the reason why we assume that *K*nowledg*e*-based *Q*uestion answer*ing* could become a CoT mentor of LLMs and name our framework as *Keqing*, with the same name of a wise character in Genshin Impact. Here we summarize the contributions of this paper as follows:

- We develop a new framework termed *Keqing* to accomplish KBQA tasks with LLMs, whose workflow mainly consists of four stages, specifically *Question Decomposition*, *Knowledge Retrieval*, *Candidate Reasoning*, and *Response Generation*, greatly improving the reliability of existing LLM's response.

- Moving beyond straightforward text-to-SQL generation, we introduce question templates as an intermediary to make it easy for LLMs to capture the logic of question decomposition, where each question template can be solved with several pre-collected logical chains.

- Distinct from constructing CoT with heuristic hand-crafted methods, we are the first to utilize the logical chains of KBQA as CoTs to guide existing LLMs to decompose complex questions into several sub-questions, which is automated and can be easily scalable.

- Experimental results show that *Keqing* can not only achieve competitive performance on recent popular benchmarks, but also become a trustworthy system to illustrate the underlying logic of answering each question, improving the interpretability of its response.

## 2 RELATED WORKS

### 2.1 RETRIEVAL-AUGMENTED LANGUAGE GENERATION

To avoid generating non-factual and out-of-data response, retrieval-augmented LMs (Mialon et al., 2023) are developed to combine elements of both retrieval-based and generation-based models to improve the quality and relevance of text generation. Existing retrieval-augmented LMs mainly rely on two types of retrievers to assess the relevance of a document to an information-seeking query, where one is the sparse retriever (Robertson et al., 2009) working with bag-of-words representations of documents and another one is the dense neural retriever (Asai et al., 2021) using dense document vectors embedded by a neural network. Moving beyond retrieving on text corpus, recent works (Li et al., 2023a) tends to explore methods for retrieving on knowledge graphs, which propose to utilize the inference capability of LLMs to directly generate executable logical chains. Distinct from these aforementioned methods, the retrieval procedure of *Keqing* adopts the form of first decomposing the problem into sub-problems and then mapping each sub-problem into logical chains, which alleviates the issue of LLMs having difficulty understanding meaning of logical chains in the form of SQL.

### 2.2 LLMs FOR KNOWLEDGE BASED QUESTION ANSWERING

Recently, large language models (LLMs), *e.g.* ChatGPT (Ouyang et al., 2022), have exhibited their potentials in precisely understanding the users' intentions after the procedure of instruction tuning and reinforcement learning from human feedback (RLHF) (Ouyang et al., 2022). However, when confronted with complex instructions or questions, *e.g.* multi-hop KBQA, most LLMs often suffer from a lack of ability to break down multi-step problems into intermediate steps before arriving at an

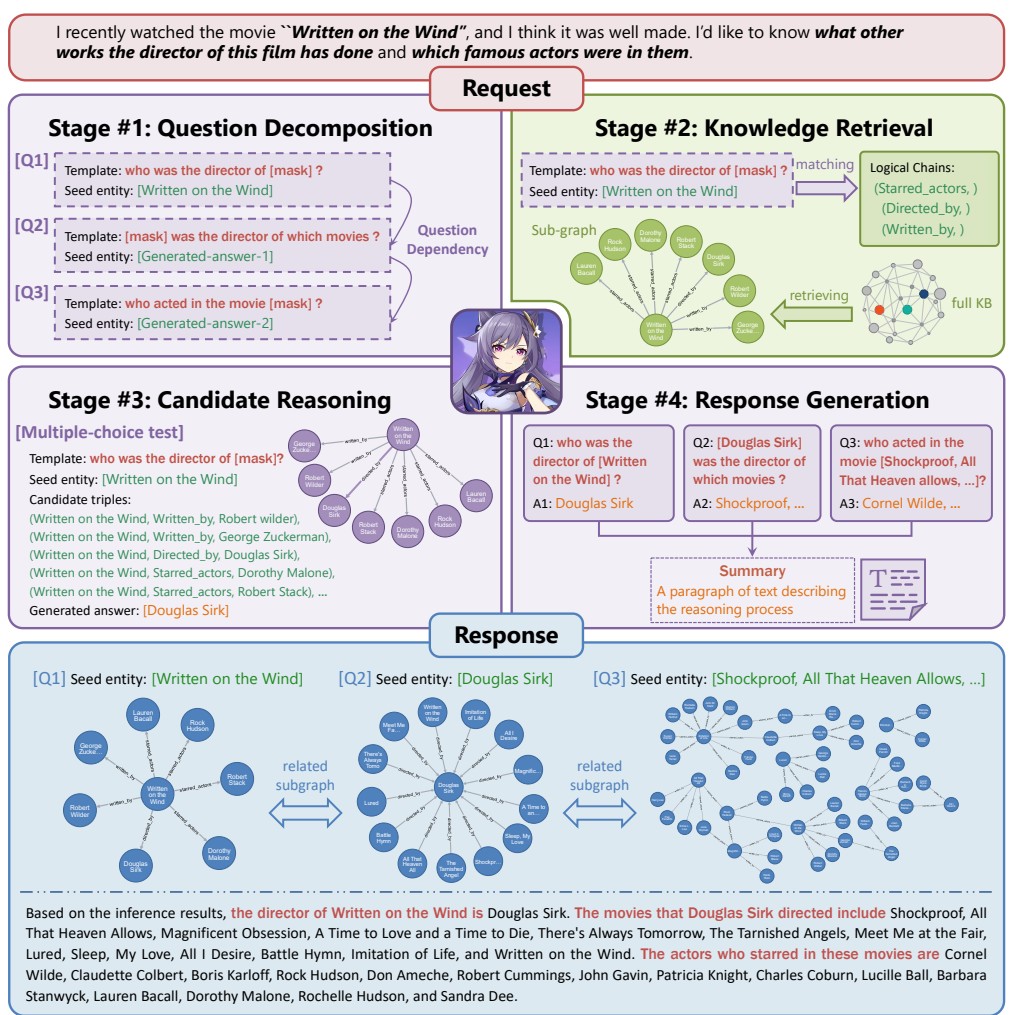

Figure 1: The workflow of *Keqing* applied for KBQA mainly consists of four stages: **#1** *Question Decomposition*: decompose a complex question into several sub-questions according to predefined question templates; **#2** *Knowledge Retrieval*: retrieve candidate entities on the knowledge graph by aligning decomposed sub-questions to pre-collected logical chains; **#3** *Candidate Reasoning*: select the correct answer from the candidate answers to solve each sub-question; **#4** *Response Generation*: generate response by summarizing multiple rounds of questions and answers.

answer, motivating recent works on chain-of-thought (CoT) prompting that heavily rely on heuristic hand-crafted algorithms (Wei et al., 2022; Kojima et al., 2022; Wei et al., 2022). Focused on the task of KBQA, distinct from previous works' conducting text-to-SQL generation with LLMs (Cheng et al., 2022; Li et al., 2023a), where the generated SQL drafts are usually not guaranteed to be executable, *Keqing* treats the logical chains on the knowledge as a mentor of CoT generation to guide LLMs to decompose complex questions into several sub-questions and then sequentially accomplish these sub-questions, where the framework is automated and can be easily scalable to large-scale datasets.

## 3 METHOD

As shown in Fig. 1, the workflow of *Keqing* mainly consists of four modules, specifically *Question Decomposition*, *Knowledge Retrieval*, *Candidate Reasoning*, and *Response Generation*, and we will introduce the technical details of each module in the following subsections.

### 3.1 DECOMPOSE COMPLEX QUESTIONS THROUGH SLOT FILLING

Under the scenario of KBQA, given a natural language query $q$, the target of KBQA is to retrieve an answer list $\mathcal{A}$ from a symbolic knowledge graph (KG) denoted as $\mathcal{K}$ for the query $q$. Assuming

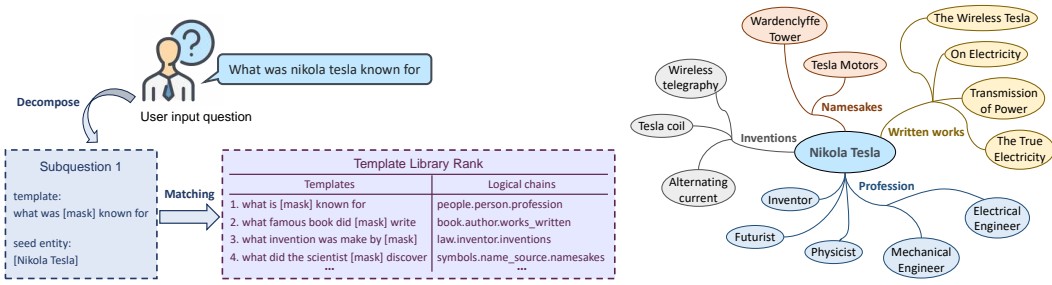

Figure 2: The pipeline of aligning decomposed sub-questions to executable logical chains on KG, where each sub-question will be mapped to a set of logical chains of top-K relevant question templates.

a training set $\mathcal{D} = \{(q_i, \mathcal{A}_i)\}_{i=1}^N$ consisting of $N$ question-answer pairs, an ideal KBQA model is supposed to learn reasoning patterns (*a.k.a.* logical chains), each of which is a subset of KG edges, from given QA pairs, and then select reasonable logical chains to deduce the answer to the query $q$.

In our consideration, the logical chains among the domain-specific knowledge graph can be naturally utilized as CoTs to guide LLMs to solve a series of complex multi-hop questions, motivating us to firstly decompose each complex question into simpler sub-questions according to the predefined question templates and then solve these sub-questions one by one with pre-collected potential logical chains. The advantages of introducing the module of *Question Decomposition* are two folds: 1) compared to the code form of SQL instructions, the text form of sub-questions are much easier to be learned by LLMs, like LLaMA (Touvron et al., 2023) and Vicuna (Chiang et al., 2023), most of whose pre-training corpus is still in text form; 2) for each question in our daily life, especially in the filed of *math* or *science*, there could be several solutions to solve the same question, where the sufficient pre-collected logical chains for each kind of question template can contribute to find multiple potential answer candidates and provide tolerance for the following reasoning procedure.

**Question Decomposition following the Chain-of-Thoughts of KBQA**

Aimed at decomposing a complex KBQA question into several simple sub-questions, a straightforward method could be directly providing exemplars in demonstration and force the LLMs to imitatively decompose the input question, following the pipeline of HuggingGPT (Shen et al., 2023). However, limited by the resource of input tokens, the exemplars in demonstration should be carefully selected to achieve a promising performance, which will cause additional resource costs and even lead to a failure due to an unsuitable exemplar selection. Thus, for the *Question Decomposition* module of *Keqing*, we decide to use LORA (Hu et al., 2021) to finetune LLMs, specifically LLaMA (Touvron et al., 2023) in our experiments, to capture the underlying mechanism of decomposing complex KBQA questions.

Formally, given a complex question (query) $q_i$ from user and a set of predefined sub-question templates $\mathcal{Q} = \{q^{(k)}\}_{k=1}^K$, the target of the *Question Decomposition* module in *Keqing* is to decompose the given query $q_i$ into $T$ sub-questions through the generation of LLMs, formulated as:

$$\{q_{i,t}\}_{t=1}^T = \textbf{LLM}(q_i), \quad q_{i,t} \in \{q^{(k)}\}_{k=1}^K, \tag{1}$$

where the training objective of each sub-question $q_{i,t}$ is to be belonged to one of $K$ predefined question templates. As the formulation of prompt and instruction shown in Table 1, taking the original question $q_i$ as the input query, LLMs are finetuned to filling the slots of sub-questions $q_{i,t}$ by generation, equipped with the seed entities and dependencies of these sub-questions.

For instance, to solve the 3-hop MetaQA question in Fig. 1, specifically "*..., what other works the director of Written on Wind has done and which famous actors were in them?*", *Keqing* is supposed to sequentially answer the following questions: 1) "*who was the director of [mask]?*", 2) "*[mask] was the director of which movies?*", and 3) "*who acted in the movie [mask]?*". Besides, *Keqing* will also automatically detect the seed entity "*Written on Wind*" and then forward it coupled with the first question "*who was the director of [mask]?*" to the following procedures to obtain the answer entities, which will be treated as the seed entities of the second question "*[mask] was the director of which movies?*" and iteratively result into the final answers according to the question dependency.

Table 1: The details of the prompt design of each module in *Keqing*. The *Execution Logs* in *Response Generation* module indicates the record of multiple rounds of questions and answers.

| Module Name | Prompt Templates |
|---|---|
| *Question Decomposition* | **Instruction:** The AI assistant can parse the user input to several subquestions: 
 **Input:** I recently watched the movie Written on the Wind, and I think it was well made. I'd like to know what other works the director of this film has done and which famous actors were in them. 
 **Output:** {"question": "who was the director of [mask]?", "id": 0, "dep": [-1], "args": {"seed_entities": ["Written on the Wind"]}}, {"question": "[mask] was the director of which movies?", "id": 1, "dep": [0], "args": "seed_entities": ["<GENERATED>-0"]}}, {"question": "who acted in the movie [mask]?", "id": 2, "dep": [1], "args": "seed_entities": "<GENERATED>-1"]}}] |
| *Candidate Reasoning* | **Instruction:** Use the following pieces of context to answer the users question. If you don't know the answer, just say that you don't know, don't try to make up an answer. 
 —————— 
 "Written on the Wind written by Robert wilder. Written on the Wind written by George Zuckerman. Written on the Wind directed by Douglas Sirk. Written on the Wind starred actors Dorothy Malone." 
 **Input:** Who was the director of Written on the Wind? 
 **Output:** ["Douglas Sirk"] |
| *Response Generation* | **Instruction:** With the task execution logs, the AI assistant needs to describe the process and inference results. Please first think cafefully and directly answer my request based on the inference results. Then please detail your workflow step by step including the used models and inference results for my request in your friendly tone. Please filter out information that is not relevant to my request. If there is nothing in the results, please tell me you can't make it. 
 **Input:** Execution Logs. 
 **Output:** Based on the inference results, the director of Written on the Wind is Douglas Sirk. The movies that Douglas Sirk directed include Shockproof, All That Heaven Allows, Magnificent Obsession, A Time to Love and a Time to Die, There's Always Tomorrow, The Tarnished Angels, Meet Me at the Fair, Lured, Sleep, My Love, All I Desire, Battle Hymn, Imitation of Life, and Written on the Wind. The actors who starred in these movies are Cornel Wilde, Claudette Colbert, Boris Karloff, Rock Hudson, Don Ameche, Robert Cummings, John Gavin, Patricia Knight, Charles Coburn, Lucille Ball, Barbara Stanwyck, Lauren Bacall, Dorothy Malone, Rochelle Hudson, and Sandra Dee. |

**Align Decomposed Sub-questions to the Logical Chains on Knowledge Graph**

Considering it is not guaranteed that the generated sub-questions will exactly match the predefined question templates during the inference phase, thus we introduce an additional template-matching procedure to fill this gap, as shown in Fig. 2. With the same notation in Eq. (1) denoting the generated sub-questions as $\{q_{i,t}\}_{t=1}^{T}$ and predefined question templates as $\{q^{(k)}\}_{k=1}^{K}$, the template-matching process aims to map each sub-question $q_{i,t}$ to its most relevant question templates, resulting in a set of logical chains to be executed on the knowledge graph for retrieving potential answer candidates.

Formally, inspired by recent works (Das et al., 2022), we propose to use RoBERTa (Liu et al., 2019), a popular variant of BERT (Devlin et al., 2018), to encode both the decomposed sub-questions $\{q_{i,t}\}_{t=1}^{T}$ and predefined questions $\{q^{(k)}\}_{k=1}^{K}$ to the same latent space, and then measure their semantic distances with cosine similarity, specifically:

$$h_{q_{i,t}} = \textbf{BERT}(q_i), h_{q^{(k)}} = \textbf{BERT}(q^{(k)}), \text{sim}(q_{i,t}, q^{(k)}) = \frac{h_{q_{i,t}}^T h_{q^{(k)}}}{||h_{q_{i,t}}||||h_{q^{(k)}}||}. \tag{2}$$

According to the obtained similarity scores, we can rank the relevance between $q_{i,t}$ and $\{q^{(k)}\}_{k=1}^{K}$, and assign the most relevant question template. We note that it is also reasonable to select multiple question templates for a single sub-question to extend the scope of retrieved answer candidates, and will investigate its influence in the following experiments.

For each question template $q^{(k)}$, we will collect a set of logical chains from KBQA dataset to try to answer this question, and the quality of the projection from question template to the set of collected logical chains will directly influence the performance of *Keqing*. The most ideal choice could be constructing the projection by human, but will be extremely time and resource consuming. Thus, in this paper, we first assign the most frequent question template to each logical chain according

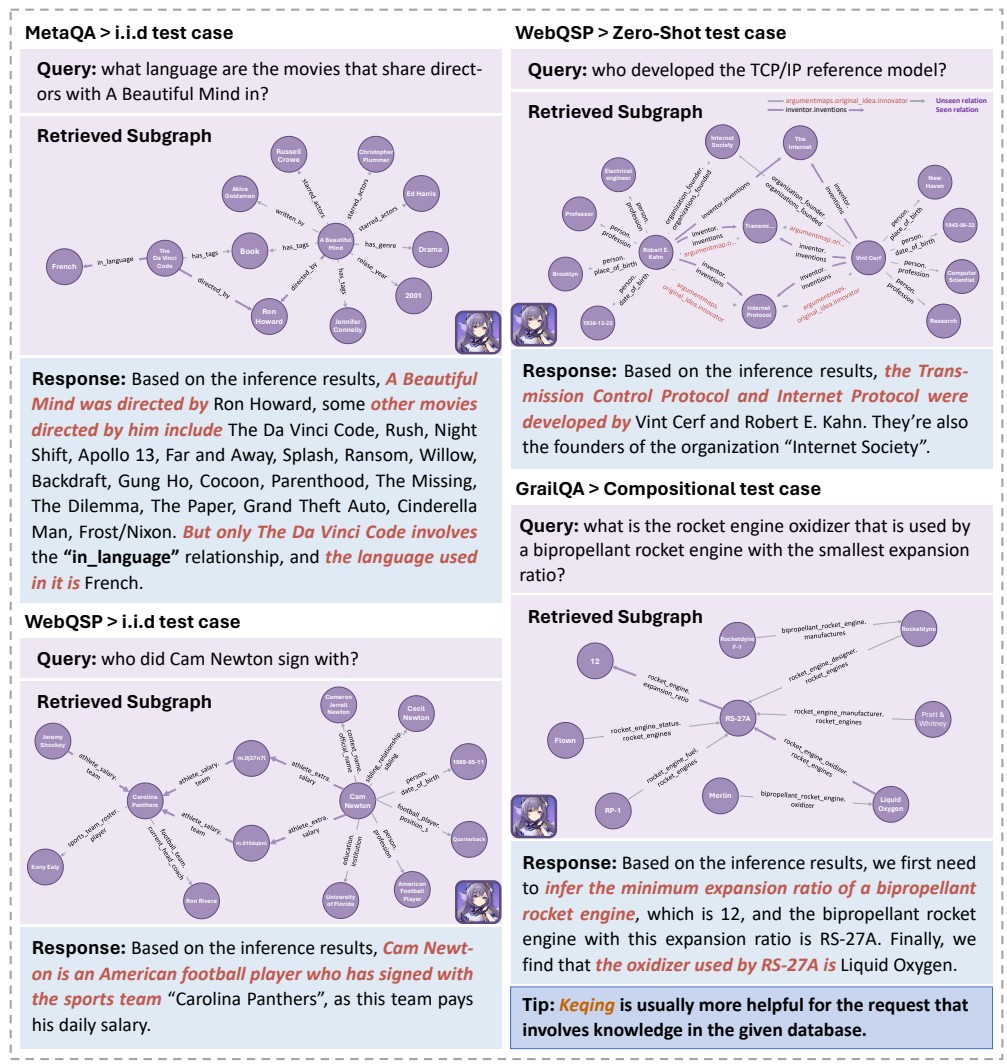

Figure 3: Case study of evaluating *Keqing* on the testing samples of various KBQA benchmarks.

the statistics in the training dataset, and then reverse the projection relationships to obtain the set of potential logical chains, denoted as $R^{(k)}$, to solve each question template $q^{(k)}$, where $R^{(k)}$ could consist of several logical chains with various lengths (not limited to 1-hop).

## 3.2 RETRIEVE CANDIDATE ENTITIES ON KNOWLEDGE GRAPH

After obtaining the seed entity and matching the decomposed sub-questions to the corresponding logical chains, the target of *Knowledge Retrieval* module is to search the answer candidates along the logical chains on the knowledge graph. Formally, given the sub-question $q_{i,t}$ marked with seed entity $s_{i,t}$ and the set of collected logical chains $R_{i,t} = \{r_{i,t}^{(l)}\}_{l=1}^{L_{i,t}}$, where each $r_{i,j}^{(l)}$ defines an executable single or multi-hop reasoning path on the knowledge graph. Starting from the seed entity $s$, we can perform logical reasoning along $r_{i,j}^{(l)}$ and obtain the resulting triplets in the following formulation:

$$(s, r, o) := (subject, relation, object), \qquad (3)$$

which represents that subject has the relation to the object, resulting in a set of triplets including optential answer candidates, denoted as $C_{i,t} = \{(s_{i,t}, r_{i,t}^{(l)}, o_{i,t}^{(l)})\}_{l=1}^{L_{i,t}}$.

Compared to traditional embedding-based knowledge retrieval methods (Karpukhin et al., 2020), the *Knowledge Retrieval* module in *Keqing* is mainly based on symbolic logical chains and can collect

answer candidates along more precise and interpretable reasoning paths, greatly reducing the resource cost of input tokens. Moreover, if there remains additional token budget left for the context input, these triplets retrieved by the embedding-based methods can be treated as a supplement to the context input, which can further improve the sufficiency of knowledge base to answer the corresponding question. In practice, we note that the triplets retrieved by DPR (Karpukhin et al., 2020) will also be included as supplementary answer candidates to broaden the knowledge retrieval results.

### 3.3 ANSWER QUESTIONS WITH RETRIEVED CANDIDATE ENTITIES

With the retrieved answer candidates $C_{i,t} = \{(s_{i,t}, r_{i,t}^{(l)}, o_{i,t}^{(l)})\}_{l=1}^{L_{i,t}}$ in hand, the target of *Candidate Reasoning* is to select the correct entities to answer the current question $q_{i,t}$, where the challenge remains how to let LLMs to understand the triplets and process the following reasoning procedure.

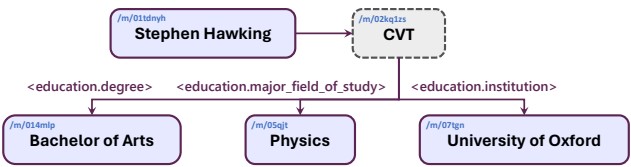

Figure 4: The *compound value types* (CVTs) of Freebase dataset, where each triplet $(s, r, o)$ will be converted to text by serializing their text surface forms.

**Formulate Retrieved Triplets to be Understood by LLMs**

In our settings, there are two distinct solutions to make the LLMs to understand the logical relationships among the retrieved triplets. The first way is to explain the composition of the triplet in the instruction part of the prompt, specifically highlight the rules: 1) the tuple format $(s, r, o)$ represents the subject $s$ has the relation $r$ to the object $o$; 2) the answer to the question should be based on given tuples and exactly consistent with the subject $s$ or object $o$. Another way is to directly convert the triplet into text using simple heuristics, such as serializing the triplet $(s, r, o)$ by concatenating the text surface forms of subject, relation and object, as shown in Fig. 4. In practice, we found that the first method is suitable for training-free LLMs, such as ChatGPT, and the second method is suitable for LLMs that requires the stage of finetuning, such as LLaMA.

**Reason Candidates to Answer Question with LLMs**

After making the LLMs understanding the formulation of triplets, given the answer candidates $C_{i,t} = \{(s_{i,t}, r_{i,t}^{(l)}, o_{i,t}^{(l)})\}_{l=1}^{L_{i,t}}$ and input question $q_{i,t}$, we will force *Keqing* to read the context by adding the prompt on the front, where the content is "use the following pieces of context to answer the users question." as shown in Table 1, and then utilize the reasoning capability of LLMs to select the correct answers, formulated as

$$C_{i,t}^* = \textbf{LLM}(q_{i,t}|C_{i,t} = \{(s_{i,t}, r_{i,t}^{(l)}, o_{i,t}^{(l)})\}_{l=1}^{L_{i,t}}), \tag{4}$$

where $C_{i,t}^*$ denotes the subset of retrieved answer candidates generated by LLMs. For the selection of LLMs to play the role of *Candidate Reasoning* module, the ideal choice should be ChatGPT, which owns excellent capability of logical reasoning to select correct answers from context and zero-shot generalization to solve unseen questions. Another solution could be to finetune an open-source LLMs, the same as *Question Decomposition* described in Section. 3.1, which would be more suitable for domain-specific KBQA.

### 3.4 GENERATE RESPONSE BY SUMMARIZING QUESTION ANSWERS

At last, after multiple rounds of questions and answers, for each complex question $q_i$, we can finally obtain the decomposed sub-questions $\{q_{i,t}\}_{t=1}^T$ and corresponding generated answers $\{C_{i,t}^*\}_{t=1}^T$, which can be treated as an execution log. To allow users to understand the logic of KBQA more intuitively, we introduce a *Response Generation* module to summarize the inference process of *Keqing*, by introducing the prompt "with the task execution logs, the AI assistant needs to describe the process and inference results..." shown in Table. 1, equipped with the execution log as input. Finally, *Keqing* can generate a comprehansive response as shown in the response part of Fig. 1.

Table 2: Performance comparison of different methods on the MetaQA benchmark (Hits@1 in percent).

| Method | MetaQA | | |
|---|---|---|---|
| | **1-hop** | **2-hop** | **3-hop** |
| KVMemNN (Miller et al., 2016) | 96.2 | 82.7 | 48.9 |
| VRN (Zhang et al., 2018) | 97.5 | 89.9 | 62.5 |
| GraftNet (Sun et al., 2018) | 97.0 | 94.8 | 77.7 |
| PullNet (Sun et al., 2019) | 97.0 | **99.9** | 91.4 |
| EmbedKGQA (Saxena et al., 2020) | 97.5 | 98.8 | 94.8 |
| NSM (He et al., 2021) | 97.2 | **99.9** | 98.9 |
| CBR-SUBG (Das et al., 2022) | 97.1 | 99.8 | 99.3 |
| ChatGPT (Jiang et al., 2023) | 61.9 | 31.0 | 43.2 |
| StructGPT (Jiang et al., 2023) | 94.2 | 93.9 | 80.2 |
| KB-BINDER (Li et al., 2023a) | 93.5 | 99.6 | 96.4 |
| *Keqing*-LLaMA (Ours) | **98.4** | **99.9** | **99.6** |

Table 3: Performance comparison of different methods on the WebQSP benchmark.

| Method | F1 |
|---|---|
| GraftNet (Sun et al., 2018) | 62.8 |
| QGG (Lan & Jiang, 2020) | 74.0 |
| ReTraCk (Chen et al., 2021b) | 71.0 |
| NSM (He et al., 2021) | 69.0 |
| CBR-SUBG (Das et al., 2022) | 72.8 |
| TIARA (Shu et al., 2022) | 76.7 |
| DecAF (Yu et al., 2022) | **78.8** |
| FlexKBQA-Codex (Li et al., 2023b) | 60.6 |
| Pangu-Codex (Gu et al., 2022) | 68.3 |
| KB-BINDER-Codex (Li et al., 2023a) | 74.4 |
| *Keqing*-LLaMA (Ours) | 69.0 |
| *Keqing*-ChatGPT (Ours) | **74.9** |

# 4 EXPERIMENTS

## 4.1 DATASETS & BASELINES

We evaluate *Keqing* on three KBQA benchmark datasets, including MetaQA (Zhang et al., 2018), WebQuestionsSP (WebQSP) (Yih et al., 2016), and GrailQA (Gu et al., 2021). Table 4 lists the statistics for the train/dev/test splits of these datasets, and more explanations about the details of the datasets can be found in Appendix A. The main competitors of *Keqing* are those KBQA systems based on existing pretrained LLMs, such as ChatGPT (Jiang et al., 2023), StructGPT (Jiang et al., 2023), Pangu (Gu et al., 2022), KB-BINDER (Li et al., 2023a), FlexK-BQA (Li et al., 2023b). More details about baselines can be found in Appendix B.

Table 4: Dataset statistics.

| Dataset | Train | Dev | Test |
|---|---|---|---|
| GrailQA | 44337 | 6763 | 13231 |
| WebQSP | 3098 | - | 1639 |
| MetaQA-1hop | 96106 | 9992 | 9947 |
| MetaQA-2hop | 118680 | 14872 | 14872 |
| MetaQA-3hop | 114196 | 14274 | 14274 |

## 4.2 IMPLANTATION DETAILS

In *Question Decomposition* module, we use LLaMA (Touvron et al., 2023) finetuned by LORA (Hu et al., 2021) to decompose each complex question into a series of sub-questions, and then use RoBERTa (Liu et al., 2019) to match each sub-question with top-K relevant question templates. For *Candidate Reasoning* module, there are two choices in our consideration as descirbed as Section. 3.3, leading to two variants named *Keqing*-LLaMA and *Keqing*-ChatGPT. Finally, we adopt ChatGPT (Ouyang et al., 2022) as *Response Generation* module to summarize the execution log.

The version of ChatGPT in *Keqing* is *gpt-3.5-turbo*, and the pretrained LLaMA can be found in Huggingface (Wolf et al., 2019). We believe the performance of *Keqing* can be further improved with more powerful LLMs, like LLaMA-2 (Touvron et al., 2023), and will include the results in the future.

## 4.3 QUALITATIVE VISUALIZATION

**Case study on various KBQA benchmarks:** To demonstrate the effectiveness of *Keqing*, we conduct a comprehensive case study that covers examples involving different levels of generalization, as shown in Fig. 3. For instance, analyzing the *i.i.d* test case from MetaQA, we can see that *Keqing* precisely breaks the input question into three simple sub-questions and finally obtains the correct answer by iteratively answering each sub-question. For the *zero-shot* test case from WebQSP, even though the gold logic chain "original_idea.innovator" has not appeared in the training set, surprisingly, *Keqing* still arrives at the right answer by matching a semantically similar logic chain "inventor.inventions". For the compositional test case from GrailQA, *Keqing* demonstrates its ability to solve combinatorial problems that did not appear in the training set by utilizing the logical chains to solve sub-questions.

## 4.4 QUANTITATIVE COMPARISON

Limited by pages, we only exhibit experimental results of MetaQA and WebQSP on the main body, as shown in Table 2, and Table 4 respectively, and leave the results of GrailQA in Appendix C. From

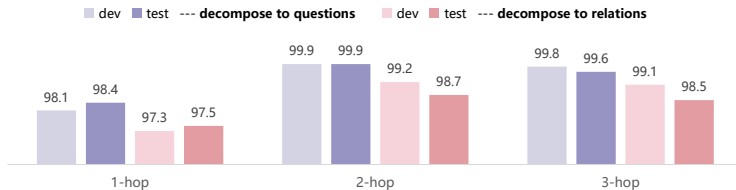

Figure 6: Performance comparison of decomposing KBQA questions into sub-questions and logical chains by finetuning LLaMA on MetaQA dataset.

the results of MetaQA, whose performance mainly depends on the quality of question decomposition, we can find that our *Keqing*-LLaMA not only outperforms traditional supervised methods but also significantly beats recent popular LLMs-based methods for KQBA, like StructGPT (Jiang et al., 2023) and KB-BINDER (Li et al., 2023a), achieving a new SOTA on this benchmark. As shown on the second block in Fig. 3, our *Keqing*-ChatGPT achieves the best performance among KBQA methods based on pretrained LLMs, demonstrating the superiority of workflow of *Keqing*, and also beats *Keqing*-LLaMA, due to the fact that the reasoning capability of ChatGPT is better than LLaMA.

## 4.5 ABLATION STUDY

For the ablation study, we mainly focus on investigating the factors that will influence the performance of *Keqing* to answer the following questions, 1) will decomposing complex problems into sub-problems using LLMs perform better than directly predicting logical chains? 2) how the number of question templates retrieved for each sub-question affects the performance of *Keqing*?

**Generate sub-questions *v.s.* generate logical chains:** As shown in Fig. 6, we conduct the performance comparison of decomposing complex questions into sub-questions and logical chains on MetaQA dataset, where the only modification is to repalce *Question Decomposition* and *Knowledge Retrieval* modules in *Keqing* with LLMs that are finetuned to directly predict logical chains. From the results, we can find that the performance of *Keqing* to accomplish KQBA tasks by generating sub-questions comprehensively outperforms the other one targeted at generating logical chains, reflecting the fact that the logic of decomposing questions in text form could be easier to be captured by pretrained LLMs than that in SQL form.

**Affect of the number of retrieved question templates:** As claimed in Section 2, *Keqing* will select top-K relevant question templates for a single sub-question to extend the scope of retrieved answer candidates, and here we investigate the influence of the number of retrieved question templates. From the results shown in Fig. 5, it is not diffiuclt to find that the KBQA performance of *Keqing* generally improves as the increase of the number of retrieved question templates, indicating that sufficient answer candidates can provide tolerance for the following procedure of answer reasoning. Moreover, this gain of performance gradually decay with the increase of the number of retrieved question templates, reflecting the fact that excessive context can cause misunderstandings of LLMs used for *Candidate Reasoning*.

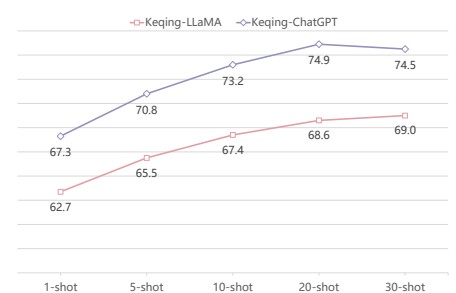

Figure 5: Performance of *Keqing* on WebQSP using different numbers of question templates to match each sub-question.

## 5 CONCLUSION

In this paper, we develop a new framework termed *Keqing* to accomplish KBQA tasks with LLMs, whose workflow mainly consists of four stages, specifically *Question Decomposition*, *Knowledge Retrieval*, *Candidate Reasoning*, and *Response Generation*, greatly improving the reliability of existing LLM's response. Moreover, the success of *Keqing* demonstrates that KBQA could be a nature CoT mentor to guide the LLM to sequentially find the answer entities of a complex question through interpretable logical chains, leading to competitive performance on KBQA tasks.

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

# A    DATASETS & PREPROCESS

**MetaQA** (Zhang et al., 2018) consists of a movie ontology derived from the WikiMovies Dataset and three sets of question-answer pairs written in different levels of difficulty. It evaluates the effectiveness in a specific domain.

**WebQSP** (Yih et al., 2016) contains questions from WebQuestions that are answerable by Freebase. It tests i.i.d. generalization on simple questions.

**GrailQA** (Gu et al., 2021) is a diverse KBQA dataset built on Freebase, covering 32,585 entities and 3,720 relations across 86 domains. It is designed to test three levels of generalization of KBQA models: I.I.D., compositional, and zero-shot.

# B    BASELINES

For the baselines in comparison, we have included the competitive methods that have a publication on the official leaderboard of each dataset and record their results from the paper directly with the same evaluation matrix. For ease of comparison, we have summarized the main thoughts of competitive baselines in the following:

**KB-BINDER** (Li et al., 2023a) is a training-free system, which for the first time, proposes to utilize the in-context learning capability of large language models (LLMs) to solve KBQA tasks. Particularly, it leverages the Codex (Chen et al., 2021a) to generate logical forms as the draft for a specific question by imitating a few demonstrations, and then grounds on the knowledge base to bind the generated draft to an executable one with BM25 score matching.

**Pangu** (Gu et al., 2022) is developed as a generic framework for grounded language understanding that capitalizes on the discriminative ability instead of the generative ability of LLMs. Specifically, Pangu consists of a symbolic agent and a neural LLM working in a concerted fashion, where the agent explores the environment to incrementally construct valid plans, and the LLM evaluates the plausibility of the candidate plans to guide the search process.

**FlexKBQA** (Li et al., 2023b) targets at leveraging automated algorithms to sample diverse programs, such as SPARQL queries, from the knowledge base, which are subsequently converted into natural language questions via LLMs. Moreover, FlexKBQA introduces an addtional execution guided self-training method to iterative leverage unlabeled user questions, which can reduce the barriers of distribution shift between synthetic data and real user questions.

# C    MORE EXPERIMENTAL RESULTS

The experimental results on GrailQA dataset have been exhibited on Table 5 and Table 6.

Table 5: Performance comparison of different methods on the GrailQA dev set.

| Method | Overall | |
|---|---|---|
| | EM | F1 |
| QGG (Lan & Jiang, 2020) | - | 36.7 |
| GloVE+Transduction (Gu et al., 2021) | 17.6 | 18.4 |
| GloVE+Ranking (Gu et al., 2021) | 39.5 | 45.1 |
| BERT+Transduction (Gu et al., 2021) | 33.3 | 36.8 |
| BERT+Ranking (Gu et al., 2021) | 50.6 | 58.0 |
| RnG-KBQA (Ye et al., 2022) | 68.8 | 74.4 |
| DecAF (Yu et al., 2022) | 68.4 | 78.8 |
| TIARA (Shu et al., 2022) | 73.0 | 78.5 |
| Pangu (Gu et al., 2022) | **73.7** | **79.9** |
| KB-BINDER (Li et al., 2023a) | 50.6 | 56.0 |
| FlexKBQA (Li et al., 2023b) | 62.8 | 69.4 |
| *Keqing*-LLaMA (Ours) | 72.5 | 77.8 |

Table 6: Performance comparison of different methods on the GrailQA dev set.

| Method | IID | | Compositional | | Zero-shot | |
|---|---|---|---|---|---|---|
| | EM | F1 | EM | F1 | EM | F1 |
| QGG (Lan & Jiang, 2020) | - | 40.5 | - | 33.0 | - | 36.6 |
| GloVE+Transduction (Gu et al., 2021) | 50.5 | 51.6 | 16.4 | 18.5 | 3.0 | 3.1 |
| GloVE+Ranking (Gu et al., 2021) | 62.2 | 67.3 | 40.0 | 47.8 | 28.9 | 33.8 |
| BERT+Transduction (Gu et al., 2021) | 51.8 | 53.9 | 31.0 | 36.0 | 25.7 | 29.3 |
| BERT+Ranking (Gu et al., 2021) | 59.9 | 67.0 | 45.5 | 53.9 | 48.6 | 55.7 |
| RnG-KBQA (Ye et al., 2022) | 86.2 | 89.0 | 63.8 | 71.2 | 63.0 | 69.2 |
| DecAF (Yu et al., 2022) | 84.8 | 89.9 | 73.4 | 81.8 | 58.6 | 72.3 |
| TIARA (Shu et al., 2022) | **88.4** | **91.2** | **66.4** | **74.8** | **73.3** | **80.7** |
| Pangu (Gu et al., 2022) | **82.6** | **87.1** | **74.9** | **81.2** | **69.1** | **76.1** |
| KB-BINDER (Li et al., 2023a) | 51.9 | 57.4 | 50.6 | 56.6 | 49.9 | 55.1 |
| FlexKBQA (Li et al., 2023b) | 71.3 | 75.8 | 59.1 | 65.4 | 60.6 | 68.3 |
| *Keqing*-LLaMA (Ours) | 80.5 | 85.6 | 73.3 | 80.1 | 67.5 | 74.7 |

To demonstrate that our approach is not only suitable for the KBQA setting but can also be extended to a broader setting, we proceed to test the efficacy of *Keqing* on the general open-domain QA task. Specifically, we focus on two multi-hop question-answering datasets, *i.e.,* HotpotQA (Yang et al., 2018) and MuSiQue-Ans (Trivedi et al., 2022b), considering that decomposition is more useful for answering complex questions that require multi-step reasoning.

HotpotQA only includes 2-hop questions and is thus relatively simple, while MuSiQue-Ans is more challenging, as it has 2,3,4-hop questions that require explicitly connected reasoning. We evaluate our method on the partial part of the two multi-hop datasets, where we use the 500 test questions for each dataset sampled by (Trivedi et al., 2022a). The results are exhibited in Table 7.

Table 7: Performance of *Keqing* on the HotpotQA and MuSiQue-Ans benchmark.

| Strategy | Method | Dataset | |
|---|---|---|---|
| | | HotpotQA | MuSiQue |
| **Few-Shot** (*In-Context Learning*) | GPT3 (Trivedi et al., 2022a) | 47.5 | 25.2 |
| | GPT3+*OneR* (Trivedi et al., 2022a) | 53.6 | 29.4 |
| | GPT3+*IRCoT* (Trivedi et al., 2022a) | 60.7 | 36.5 |
| **Zero-Shot** (*No Demonstration*) | ChatGPT | 45.7 | 24.1 |
| | ChatGPT+*OneR* | 55.3 | 30.6 |
| | ChatGPT+*decompose* (**ours**) | 54.5 | 32.4 |
| | ChatGPT+*decompose*+*OneR* (**ours**) | **62.8** | **38.9** |

In Table 7, the results of *few-shot* strategy are inherited from (Trivedi et al., 2022a), where they send demonstration examples along with the query question to GPT3 (`code-davinci-002`) for requesting the answer. For open-domain QA tasks, the retrieved context is also typically sent to GPT3 to generate the answer. One-step Retriever (OneR) directly uses the question as a query to retrieve K paragraphs, with BM25 (Robertson et al., 2009) implemented in Elasticsearch as the base retriever. Interleaving Retrieval with Chain-of-Thought (IRCoT) is the approach proposed by Trivedi et al., which interleaves CoT generation and knowledge retrieval steps to guide more effective retrieval.

While we conduct additional experiments under a more challenging *zero-shot* setting, where only the query question is sent to ChatGPT (`gpt-3.5-turbo`) to generate the answer directly. We also considered asking ChatGPT to generate answers based on the context retrieved using the same One-Step Retriever. And our approach is mainly embodied in designing a prompt that encourages ChatGPT to first break down the query question into several simpler sub-questions and then solve these sub-questions sequentially to obtain the final correct answer. Note that when combined with OneR, our method uses OneR once for each of the decomposed sub-questions respectively to derive a more matching context. The results in Table 7 suggest that our approach leads to a substantial performance gain by simply taking one more step of decomposition. And the highest F1 score under

the difficult *zero-shot* setting is even better than that of IRCoT in a moderately easy *few-shot* setting. We believe this is convincing evidence of the wide applicability of our approach.

## D  RUNTIME AND MEMORY COMPLEXITY

As presented in Figure 1, the workflow of *Keqing* mainly consists of four stages, where **#1** *Question Decomposition*, **#3** *Candidate Reasoning*, and **#4** *Response Generation* are all performed with the powerful capabilities of LLMs, while **#2** *Knowledge Retrieval* is a self-contained module that serves the purpose of searching for facts relevant to each sub-question from the given KB, which can be incorporated into any existing advanced retrieval technique.

Although we can use off-the-shelf LLMs to complete *Question Decomposition* and *Candidate Reasoning*, we instead employed a fine-tuned LLM in our experiments to achieve better performance. Concretely, we chose to train the LLaMA model with 7 billion parameters (LLaMA-7B) using a parameter-efficient fine-tuning technique, *i.e.*, LoRA (Hu et al., 2021), which we found to achieve reasonably good performance, finished on two NVIDIA Quadro RTX 8000 graphics cards with 48G memory for each. The detailed information about runtime and memory usage are listed in Table 8.

Table 8: Basic statistics of the required memory of GPUs, tuning time, batch size (denoted as BS) per device using LoRA tuning.

| Stage | Dateset | Training examples | LoRA BS | GPU usage | Training epoch | Training time |
|---|---|---|---|---|---|---|
| *Question Decomposition* | MetaQA | 32927 | 64 | 16.5G | 3 | 9.5h |
| | WebQSP | 3098 | 16 | 27.8G | 10 | 11.8h |
| | GrailQA | 44337 | 16 | 28.2G | 3 | 13.2h |
| *Candidate Reasoning* | MetaQA | 24695 | 16 | 13.3G | 3 | 4.5h |
| | WebQSP | 3098 | 4 | 23.7G | 10 | 5.7h |
| | GrailQA | 30496 | 4 | 23.8G | 3 | 7.6h |

## E  FAILURE EXAMPLES

In this section, we further analyzed the potential causes leading to *Keqing*'s failure to produce the correct answer, which can be roughly categorized into three cases: *i*) the LLM failed to decompose the original question properly; *ii*) the retrieved knowledge does not contain the correct answer to the question; *iii*) the LLM failed to extract the right answer from the given context. In the following, we display a representative example of each of the three failures.

---

**Decomposition Failure**

**Question:** what language is used by the file with top of the pops as the main song?

**Gold answer:**
[{"question_text": "which file has [mask] as the main song?", "question_id": 0, "dep": [-1], "seed_entities": ["Top of the Pops"]}, {"question_text": "what is the language used by [mask]?", "question_id": 1, "dep": [0], "seed_entities": ["<ANSWER>-0"]}]

**Generated response:**
[{"question_text": "which language is used by the program [mask]?", "question_id": 0, "dep": [-1], "seed_entities": ["Top of the Pops"]}, {"question_text": "what is the language dialect of [mask]?", "question_id": 1, "dep": [0], "seed_entities": ["<ANSWER>-0"]}]

---

**Retrieval Failure**

**Question:** what is the australian dollar called?
**Gold answer:** ["AUD"]

**Retrieved context:** The Australian topic notable types Newspaper, and newspaper owner News Corp Australia. News Corp Australia topic notable types Business Operation, and business operation industry Mass media. Qantas topic notable types Airline. Qantas phone sandbox product or service Qantas Holidays and Flight Booking, and phone sandbox service location Australia and Canada, and organization founders Fergus McMaster and Hudson Fysh. Qantas airline airport presence airport Perth Airport, and airline airport presence cities served Broome, Brisbane, Cairns, Darwin, Hong Kong, Singapore, Sydney, Tokyo.

**Generated response:** There is no relevant information in the given text about what the Australian dollar is called.

**Reasoning Failure**

**Question**: what countries are part of the uk?
**Gold answer:** ["Wales", "Northern Ireland", "Scotland", "England"]

**Retrieved context**: United Kingdom partial containment relationship partially contains Euroregion East Sussex/Seine Maritime/Somme. United Kingdom adjoining relationship adjoins Republic of Ireland and France. United Kingdom administrative area administrative children England, Northern Ireland, Scotland, and Wales. United Kingdom, with Dependencies and Territories administrative area administrative children United Kingdom, and location contains London Borough of Merton and Greater London. United Kingdom location contains East Anglia, North East Lincolnshire, and Parton. United Kingdom organization membership organization Organisation for Economic Co-operation and Development.

**Generated response:** The text does not provide any information about the countries that are part of the UK.

