# OpenReview forum: "Keqing: Knowledge-based Question Answering is A Nature Chain-of-Thought mentor of LLMs"
_ICLR.cc/2024/Conference — Submitted to ICLR 2024_

### Official Review · Reviewer_YEjv · 2023-10-24

**Soundness:** 2 fair
**Presentation:** 4 excellent
**Contribution:** 2 fair
**Rating:** 5
**Confidence:** 3

**Summary:**

The authors propose a new method of Question-Answering by utilising a combination of a Large Language Model and a Knowledge Base. The primary technique is to decompose a question into a set of simple questions, retrieve candidate answers for each simple question from Knowledge Base, and then select one candidate answer as the answer to a simple question, finally integrate all these simple question+answer into a text. On a number of benchmark datasets, this method reaches the SOTA performance. The authors claim the proposed method can perform better in the future.

**Strengths:**

This paper is very well written and easy to read. The authors applied the Chain-of-Thought (CoT)  idea to general question-answering, and very well motivated the whole design using existing techniques and datasets. This is beautiful.

**Weaknesses:**

At the methodological level, this paper is within the paradigm of the art of alchemy, in which authors demonstrated professional and proficient skills. The processes of question decomposition, selection, as well as the reasoning of answers, are all black-boxes. This might answer the question why this method has not outperformed the SOTA performance.

A small mistake is that Authors forgot to remove the reference to the Appendix, which resulted in Appendix ?? everywhere in the text.

**Questions:**

In Section 4.2, there is a sentence "we believe the performance of Keqing can be further improved with more powerful LLMs, like LLmMA-2", and will include the results in the future. How is the related with the primary method by Question-decomposition using Knowledge bases?

What if applying this method for the CoT logic reasoning? Could Keqing outperforms the SOTA level?

Can you provide one error case to illustrate the limitation of the current Keqing system?

---

> ### Author Response · Authors · 2023-11-22
>
> We appreciate your constructive comments and feedback. The weaknesses and questions have been addressed below.
>
> W1: At the methodological level, this paper is within the paradigm of the art of alchemy, in which authors demonstrated professional and proficient skills. The processes of question decomposition, selection, as well as the reasoning of answers, are all black-boxes. This might answer the question why this method has not outperformed the SOTA performance.
>
> A1: We agree with your point that LLM-based models are almost all within the paradigm of the art of alchemy. However, we also need to highlight existing has already demonstrated a certain degree of interpretability and generalization. For instance, the CoT generated by question decomposition module of Keqing can be used to interpret the logic of multi-hop question answering, which can help users to understand the process of answer generation and determine whether to adopt this answer.
>
> For the performance, we believe that our performance can be further improved with more powerful LLMs and we will included their results in our next iteration of revision.
>
> W2: A small mistake is that Authors forgot to remove the reference to the Appendix, which resulted in Appendix ?? everywhere in the text.
> A2: Thanks for pointing out this typo, we have fixed it in the updated version of the paper.
>
> Q1: In Section 4.2, there is a sentence "we believe the performance of Keqing can be further improved with more powerful LLMs, like LLmMA-2", and will include the results in the future. How is the related with the primary method by Question-decomposition using Knowledge bases?
>
> A1: Since our experiments finetuning LLMs to finish question decomposition, and we currently use a finetuned LLaMA-7B model to assist decompose input questions, where the results of question decomposition will heavily influence the performance of generated answers, and we believe this stage can be improved by finetuning more powerful LLMs.
>
> Q2: What if applying this method for the CoT logic reasoning? Could Keqing outperforms the SOTA level?
> A2: Actually, the decomposed sub-questions according to predefined question template can be already treated as the CoT for multi-hop question answering. And we are very willing to include comparison of CoT logic reasoning in our feature work.
>
> Q3: Can you provide one error case to illustrate the limitation of the current Keqing system?
>
> A3: Thanks for your suggestions, we have further analyzed the potential causes leading to Keqing’s failure and presented three representative failure examples (decomposition failed; retrieval failed; reasoning failed) in Appendix. Please refer to Appendix E in the updated version of the paper, new added content is marked in blue.

---

> > ### Comment · Reviewer_YEjv · 2023-11-22
> >
> > "However, we also need to highlight existing has already demonstrated a certain degree of interpretability and generalization. For instance, the CoT generated by question decomposition module of Keqing can be used to interpret the logic of multi-hop question answering"
> >
> > You are right, CoT introduces the structure into the reasoning of LLMs by decomposing a complex task into several intermediate steps. But these intermediate steps remain black boxes.  The "certain degree of interpretability" are built upon these black boxes. The real challenge is not how to decompose, but how to solve these intermediate steps. CoT is only a fig leaf.

---

> ### Author Response · Authors · 2023-11-23
> **Further response to Reviewer YEjv**
>
> We thank the reviewer for joining the discussion!
>
> We respectfully disagree with you that the real challenge is how to solve these intermediate steps. Since for the Knowlesge Base Question Answering (KBQA) task, even traditional methods perform well on the one-hop datasets, because one only needs to find the corresponding fact (typically a triplet ($s, r, o$)) in the given KB to answer one-hop questions. While the main challenge lies in how to solve those diffcuclt questions that requires multi-steps of reasoning, and with the decomposition module in our framework, each intermediate step becomes answering a one-hop sub-question, which is finished using LLMs in our approach. As we believe LLMs are powerful enough to automatically extract correct answers from the retrieved context for one-hop sub-questions, therefore, decomposition has turned out to be an effective strategy to solve the complex multi-hop questions for KBQA task.
>
> Best regards.

---

> > ### Comment · Reviewer_YEjv · 2023-11-23
> >
> > Thanks for the quick feedback. I am afraid you assumed or expected too much for LLMs. Even LLMs appear to be able to extract "correct" answers for one-hop sub-questions, LLMs are never sure or know their retrieved answers are "correct" or not. When they retrieved "correct" answers, their explanation, if you ask, may not be correct (consistent with the answer).

---

> > > ### Author Response · Authors · 2023-11-23
> > > **Further  discussion**
> > >
> > > We appreciate your kind reminder that we may expect too much for LLMs, given that the reliability of LLMs remains an open problem. Especially, the emergence of hallucination phenomena has raised concerns that they do not always speak on good grounds (*e.g.*, their explanations may not be consistent with the answers). However, this is a common unresolved issue with LLMs that should not be a reason to prevent people from applying them. Moreover, our main contribution lies in developing an effective framework to complete complex KBQA tasks, and the candidate reasoning stage can also be finished using established GNN-based approaches if you consider LLMs are not reliable.

---

> > > > ### Comment · Reviewer_YEjv · 2023-11-23
> > > >
> > > > Agree. Then, you need to describe the motivation why you use them, and may also need to compare with other methods in experiments. All seem not sufficiently addressed in this submission.

---

> > > > > ### Author Response · Authors · 2023-11-23
> > > > >
> > > > > We sincerely appreciate your active discussions with us and timely feedback. We believe that your valuable suggestions will significantly help us to improve the quality of our paper in the subsequent revision. Thank you again!
> > > > >
> > > > > Best Regards.

---

### Official Review · Reviewer_6eaX · 2023-10-31

**Soundness:** 3 good
**Presentation:** 3 good
**Contribution:** 2 fair
**Rating:** 5
**Confidence:** 4

**Summary:**

This paper presents a model to assist LLMs, such as ChatGPT, to retrieve question-related structured information on the knowledge graph, and demonstrates that Knowledge-based question answering (Keqing) could be a nature Chain-of-Thought (CoT) mentor to guide the LLM to sequentially find the answer entities of a complex question through interpretable logical chains. Specifically, the workflow of Keqing will execute decomposing a complex question according to predefined templates, retrieving candidate entities on knowledge graph, reasoning answers of sub-questions, and finally generating response with reasoning paths.

**Strengths:**

Experiments on one-hop, two-hop, and three-hops are interesting, and the baseline methods compared against seem to be comprehensive, with good experiment results demonstrated.

**Weaknesses:**

1. Recent developments in question answering also consider utilizing graph neural network methods e.g.,
Question-Answer Sentence Graph for Joint Modeling Answer Selection. In Proceedings of the 17th Conference of the European Chapter of the Association for Computational Linguistics, pages 968–979, Dubrovnik, Croatia. Association for Computational Linguistics.

2. I have a concern regarding the novelty of the approach. This work simply uses RoBERTa-based similarity scores and DPR-based knowledge augmentation, both works which have already been proposed before. The authors need to better highlight their contributions and why they consider it original work.

**Questions:**

Can the authors also illustrate the runtime and memory complexity of their work, as it is highly dependent upon LLMs which incur large runtime for finetuning?

---

> ### Author Response · Authors · 2023-11-22
>
> We appreciate your constructive comments and feedback. The weaknesses and questions have been addressed below.
>
> W1: Recent developments in question answering also consider utilizing graph neural network methods e.g., Question-Answer Sentence Graph for Joint Modeling Answer Selection. In Proceedings of the 17th Conference of the European Chapter of the Association for Computational Linguistics, pages 968–979, Dubrovnik, Croatia. Association for Computational Linguistics.
>
> A1: Thanks for notification, and we have cited this paper in our revision already.
> We know there are a lot of outstanding KBQA framework built on Graph Neural Network or some other sophisticated networks. However, compared to the performance, the explainability and generalizability should be also considered as the metric of measuring KBQA system. Compared to traditional KBQA systems, Keqing can intuitively explain the logic of generating multi-hop answers, leading to a trustworthy LLM-based KBQA system, which is currently a popular research direction. Meanwhile, Keqing also demonstrated strong generalization abilities by leveraging the reasoning capabilities of LLMs, and we believe its performance can be further improved as the increase of capabilities of LLMs."
>
>
> W2: I have a concern regarding the novelty of the approach. This work simply uses RoBERTa-based similarity scores and DPR-based knowledge augmentation, both works which have already been proposed before. The authors need to better highlight their contributions and why they consider it original work.
>
> A2: We acknowledge that question decomposition and candidate reasoning are not fresh ideas in the Q&A domain, however, to the best of our knowledge, this is indeed the inaugural endeavor to build a pipeline integrating these functions through the magic of LLMs. With respect to novelty, we would like to emphasize that we have developed an effective framework that can automate the complete process of knowledge-based question answering with as little extra effort as possible. Note that our pipeline can be training-free with least effort by using off-the-shelf LLMs to finish question decomposition and candidate reasoning while maintaining a decent performance, although a little fine-tuning effort can lead to better performance.
> Distinct from traditional KBQA systems,
>
> In addition, our approach also differs from those recent KBQA systems based on LLMs that typically generate one-step symbolic expressions for the query question leveraging the in-context learning capabilities of LLMs, which is error-prone and lacks transparency.
>
>
> Q1: Can the authors also illustrate the runtime and memory complexity of their work, as it is highly dependent upon LLMs which incur large runtime for finetuning?
>
> A1: Thanks for your valuable advice, we have included the additional information about the runtime and memory complexity of our method in Table.8 of Appendix.
> More details can found in Appendix D in the updated version of the paper, where the new added content has been marked in blue.
>
> Specifically, although we can use off-the-shelf LLMs to complete Question Decomposition and Candidate Reasoning, we instead employed a fine-tuned LLM in our experiments to achieve better performance. In practice, we chose to finetune the LLaMA model with 7 billion parameters (LLaMA-7B) using a parameter-efficient fine-tuning technique, i.e., LoRA, which we found to achieve reasonably good performance, finished on two NVIDIA Quadro RTX 8000 graphics cards with 48G memory for each. Thus, the finetuning of Keqing won’t cost too much resources of computation and memory storage.

---

> > ### Comment · Reviewer_6eaX · 2023-11-23
> >
> > Thanks to the authors for providing a response to the questions raised. However, overall, your answers to the above questions still leave us with significant concerns, so I cannot raise my score. However, it seems that the paper is promising and with appropriate revision has good potential in another future venue.

---

> > > ### Author Response · Authors · 2023-11-23
> > > **Thanks for your reply**
> > >
> > > Thank you again for your further comments. Please let us know which of your significant concerns or any questions have not been addressed, we are more than happy to clarify more about our paper and discuss it further with you.
> > >
> > > Best regards.

---

### Official Review · Reviewer_jydE · 2023-11-01

**Soundness:** 2 fair
**Presentation:** 3 good
**Contribution:** 2 fair
**Rating:** 3
**Confidence:** 3

**Summary:**

This paper presents a knowledge graph question answering using LLMs using chain of thought prompting of LLMs. Authors propose a 4 step process to process KBQA using LLMs namely Question Decomposition, Knowledge, Retrieval, Candidate Reasoning, and Response Generation. Authors try to show that using these steps to prompt LLMs can generate better response than text-SQL or structured query generation. This is demonstrated through experiments with few KBQA datasets and openly available LLMs.

**Strengths:**

Experimental results showing the effectiveness of the approach on two KBQA benchmarks.
Adapting question logical forms to aide chain of thought prompting in LLMS for KBQA.

**Weaknesses:**

Question decomposition to aid better performance is studied in the literature through BREAK paper etc. Only difference I see is just applying or solving some of those problems using LLMs and stitching the pipelines together. Not sure about the novelty of the overall approach.
Authors claim KBQA can be a nature guide to help LLMs in CoT prompting. I don't see how this can be transferred to other settings like lets say normal Open-domain QA using LLMs. Any results to show that these method can aid in solving open domain QA as well? Applicability of methods proposed methods beyond KBQA setting.
Writing can be improved and Appendix reference missing consistently across the paper.

**Questions:**

1. How are the answers for KBQA are extracted for final F1 measure ? since LLMs generate free text, what method is used to extract the final answer from the LLM response ?

---

> ### Author Response · Authors · 2023-11-22
>
> W1. Not sure about the novelty of the overall approach.
>
> A: We acknowledge that question decomposition and candidate reasoning are not fresh ideas in the Q&A domain, however, to the best of our knowledge, this is indeed the inaugural endeavor to build a pipeline integrating these functions through the magic of LLMs. With respect to novelty, we would like to emphasize that we have developed an effective framework that can automate the complete process of knowledge-based question answering with as little extra effort as possible. Note that our pipeline can be training-free with least effort by using off-the-shelf LLMs to finish question decomposition and candidate reasoning while maintaining a decent performance, although a little fine-tuning effort can lead to better performance.
> Distinct from traditional KBQA systems,
>
> In addition, our approach also differs from those recent KBQA systems based on LLMs that typically generate one-step symbolic expressions for the query question leveraging the in-context learning capabilities of LLMs, which is error-prone and lacks transparency
>
>
> W2. Any results to show that these method can aid in solving open domain QA as well? Applicability of methods proposed methods beyond KBQA setting.
>
> A: To demonstrate that our approach is not only suitable for the KBQA setting but can also be extended to a broader setting, we proceed to test the efficacy of Keqing on the general open-domain QA task. As the experimental results on two multi-hop question-answering datasets, HotpotQA and MuSiQue-Ans, our proposed Keqing achieved new SOTA performance on both these two open-domain QA datasets, demonstrating the effectiveness of our framework design. More experimental details can be found in Appendix C in the updated version of the paper, where new added content has been marked in blue.
>
>
> W3. Writing can be improved and Appendix reference missing consistently across the paper.
> A: Thanks for your suggestion. We have completed the missing Appendix reference in the updated paper and we will endeavor to improve the writing of the final manuscript.
>
>
> Q1. How are the answers for KBQA are extracted for final F1 measure? since LLMs generate free text, what method is used to extract the final answer from the LLM response?
>
> A: Thanks for your question. Indeed, this is related to the prompt we used during the “Candidate Reasoning” stage, where we intentionally included an instruction for the LLMs’ response format. Specifically, the prompt is as follows.
>
> You will be provided with a block of text and a question, and your task is to extract all the answer to the question from the given text. Do not generate duplicate answers and do not make up answers if there is no relevant information in the text. The ouput must be in a strict List format.
>
> Therefore, with this prompt, we illustrate an input-output example.
> Input:
> Text: {A paragraph describing the facts retrieved from the Knowledge Graph}\n Question: what does jamaican people speak?
> Output:
> ["Jamaican English", "Jamaican Creole English Language"]
>
> Then we compute the F1 score by comparing the output list and the gold answer list. Occasionally the LLM’s responses are not in List format, for those free-text outputs, we manually convert them to lists containing only the answer phrases so that F1 score can be obtained in the same way as above.

---

### Official Review · Reviewer_eCHK · 2023-11-02

**Soundness:** 3 good
**Presentation:** 3 good
**Contribution:** 2 fair
**Rating:** 5
**Confidence:** 4

**Summary:**

The paper introduces "Keqing", a groundbreaking framework aimed at amplifying the performance of Large Language Models (LLMs) in knowledge-based question answering (KBQA). While LLMs, such as ChatGPT, have demonstrated notable proficiency in various NLP tasks, they occasionally generate incorrect or nonsensical responses, particularly when faced with questions that exceed their training data's scope. To counter this, Keqing incorporates an IR module to extract structured information from a knowledge graph, systematically guiding the LLM to answer intricate questions. This methodology not only bolsters the trustworthiness of the LLM's answers but also ensures that these responses are interpretable.

**Strengths:**

1.The comprehensive four-stage workflow (decomposition, retrieval, reasoning, and response generation) offers a systematic approach to knowledge-based question answering.

2. The framework guarantees that the produced answers are not just accurate but also transparent, revealing the logical journey leading to the conclusion.

3. Experiments conducted on GrailQA, WebQ, and MetaQA validate the effectiveness of the framework.

**Weaknesses:**

1. The title is somewhat misleading, obscuring the paper's main contribution. Given that the paper primarily centers on a novel framework integrating an IR module to derive structured data from a knowledge graph, the connection between this pipeline and CoT remains unclear.

2. The paper's novelty in comparison to traditional KBQA systems is ambiguous. While elements like question decomposition and candidate reasoning aren't new to the field, it's uncertain whether this is the first instance of such a pipeline being employed with LLMs.

3. The model's performance in a few-shot scenario appears to lag behind state-of-the-art fine-tuned models, such as Decaf. It would be beneficial to pinpoint the reason for this shortfall or determine which step in the process contributes to this gap.

**Questions:**

1. The title of the paper suggests a focus on the CoT mentor, but the main content seems to be centered around a framework that integrates an IR module with a knowledge graph. Could you clarify the relationship between this pipeline and the concept of CoT?

2. In terms of originality, how does the proposed framework distinguish itself from traditional KBQA systems? Specifically, while aspects like question decomposition and candidate reasoning are familiar in the literature, is this the inaugural application of such a pipeline using LLMs?

3. The results indicate that the model's performance in a few-shot learning scenario is not on par with state-of-the-art models like Decaf. Could you shed light on the reasons behind this disparity? Which part of the framework or which specific stage might be contributing to this performance gap?

---

> ### Author Response · Authors · 2023-11-22
>
> Thanks, we appreciate your constructive comments and feedback. The weaknesses and questions have been addressed below.
>
> W1 and Q1: The title of the paper suggests a focus on the CoT mentor, but the main content seems to be centered around a framework that integrates an IR module with a knowledge graph. Could you clarify the relationship between this pipeline and the concept of CoT?
> A1: Thanks, it is correct that main contribution of work is to develop a novel framework to integrate an IR module to retrieve knowledge stored on graph structured datasets to answer the input user question. During the procedure of knowledge retrieval, we will first use LLM to decompose the user question into a series of sub-questions according to predefined question templates, and them as CoT to gradually answer the user question. Considering each question template in a CoT can be solved with logical chains on the knowledge graph, we propose the title ``KNOWLEDGE-BASED QUESTION ANSWERING IS A NATURE CHAIN-OF-THOUGHT MENTOR OF LLMS’’
>
> W2 and Q2: In terms of originality, how does the proposed framework distinguish itself from traditional KBQA systems? Specifically, while aspects like question decomposition and candidate reasoning are familiar in the literature, is this the inaugural application of such a pipeline using LLMs?
> A2: To the best of our knowledge, Keqing is indeed the first instance of KBQA framework to integrate the modules of Question Decomposition, Knowledge Retrieval, Candidate Reasoning, and Response Generation into one entity, which is purely deployed with LLMs.
>
> Distinct from traditional KBQA systems that either rely on laborious rule-based algorithms to achieve question decomposition [1] or individually train a graph neural network to perform candidate reasoning [2], our framework is designed to automate the whole process with as little extra effort as possible. Note that our pipeline can be training-free with least effort by using off-the-shelf LLMs to finish question decomposition and candidate reasoning while maintaining a decent performance, although using finetuned LLMs perform better. In addition, our approach also differs from those recent KBQA systems based on LLMs that typically generate one-step symbolic expressions for the query question leveraging the in-context learning capabilities of LLMs, which is error-prone and lacks transparency.
>
> W3 and Q3: The results indicate that the model's performance in a few-shot learning scenario is not on par with state-of-the-art models like Decaf. Could you shed light on the reasons behind this disparity? Which part of the framework or which specific stage might be contributing to this performance gap?
> A3: Actually, our framework is a zero-shot KBQA system rather than based on few-shot learning. Because either in the question decomposition stage or in the candidate reasoning stage, our method do not rely on imitating a few demonstrations to generate responses, i.e., we do not follow the paradigm of in-context learning. And in this respect we are consistent with FlexKBQA[3] rather than KB-BINDER[4].
>
> As for the performance gap with state-of-the-art models, we assume it is caused by the capability of generalizing of Llama during question decomposition, because the answers can hardly be correct as long as any part of generated CoT is wrong. And conviction is that the performance of Keqing can be further improved with more powerful pretrained LLMs.
>
> [1] Zheng et al., Question answering over knowledge graphs: question understanding via template decomposition. In Proceedings of the VLDB Endowment, 2018.
> [2] Das et al., Knowledge base question answering by case-based reasoning over subgraphs. In ICML, 2022.
> [3] Li et al., Flexkbqa: A flexible llm-powered framework for few-shot knowledge base question answering. In arXiv preprint, 2023.
> [4] Li et al., Few-shot in-context learning for knowledge base question answering. In arXiv preprint, 2023.

---

### Meta-Review · Area_Chair_rboq · 2023-12-09

**Metareview:**

The paper "Keqing" proposes a novel framework for enhancing Large Language Models (LLMs) in knowledge-based question answering (KBQA). The four-stage workflow includes question decomposition, knowledge retrieval, candidate reasoning, and response generation. The authors argue that this approach, termed Chain-of-Thought (CoT) mentoring, improves LLM performance on KBQA tasks. Experimental results on GrailQA, WebQ, and MetaQA datasets are presented to validate the effectiveness of the framework. However, reviewers have expressed concerns on the soundness, contribution, and presentation of the paper.

Strengths: The comprehensive four-stage workflow provides a systematic approach to KBQA, ensuring accuracy and transparency in answers. The experimental results on various KBQA benchmarks demonstrate the effectiveness of the proposed framework. The incorporation of CoT in the KBQA process is considered a well-motivated and innovative aspect of the paper.

Weaknesses: The novelty of the approach is questioned because of the lack of clarity on how it differs from existing KBQA systems. Some reviewers express uncertainty about the method's applicability beyond KBQA settings, such as in open-domain QA using LLMs; and some expressed concerns about the black-box nature of the question decomposition, selection, and reasoning processes which might hinder it from outperforming SOTA performance.

**Justification For Why Not Higher Score:**

While the paper introduces an intriguing CoT-guided KBQA framework and provides evidence of its effectiveness, concerns about novelty, clarity, and transparency in the approach need to be addressed for a clear contribution to the field. Clarifying the positioning of the work in contrast to the literature, and addressing concerns about the black-box nature of the method's processes could enhance the paper's impact.

**Justification For Why Not Lower Score:**

n/a

---

### Decision · Program_Chairs · 2024-01-16

Reject